# Neoexpression of *JUNO* in Oral Tumors Is Accompanied with the Complete Suppression of Four Other Genes and Suggests the Application of New Biomarker Tools

**DOI:** 10.3390/jpm12030494

**Published:** 2022-03-18

**Authors:** Dominik Kraus, Simone Weider, Rainer Probstmeier, Jochen Winter

**Affiliations:** 1Department of Prosthodontics, Preclinical Education and Material Sciences, University Hospital, Faculty of Medicine, University of Bonn, Welschnonnenstr. 17, 53111 Bonn, Germany; dominik.kraus@ukbonn.de; 2Department of Oral & Maxillofacial Plastic Surgery, University Hospital, Faculty of Medicine, University of Bonn, Venusberg-Campus 1, 53127 Bonn, Germany; simone.weider@ukbonn.de; 3Neuro- and Tumor Cell Biology Group, Department of Nuclear Medicine, University Hospital, Faculty of Medicine, University of Bonn, Venusberg-Campus 1, 53127 Bonn, Germany; r.probstmeier@gmx.net; 4Oral Cell and Tumor Biology Group, Department of Periodontology, Operative and Preventive Dentistry, University Hospital, Faculty of Medicine, University of Bonn, Welschnonnenstr. 17, 53111 Bonn, Germany

**Keywords:** novel tumor markers, neoexpressed genes, OSCC, molecular pathology

## Abstract

Background. Our study describes the neoexpression (Juno) and suppression (catsperD, dysferlin, Fer1L5 and otoferlin) of selected genes in oral squamous cell carcinomas (OSCCs). As the expression pattern of these genes allows a “yes” or “no” statement by exhibiting an inverse expression pattern in malignant versus benign tissues, they represent potential biomarkers for the characterization of oral malignancies, particularly OSCCs. Methods. Differential expression analyses of selected genes of interest were examined by quantitative PCR of oral cancer tissues compared to normal. Results. Five candidates out of initially nine genes were examined, demonstrating Juno as a putative new tumor marker selectively expressed in OSCCs. Interestingly, the expression of four other genes in benign tissues was completely repressed in tumor tissues with a specificity and sensitivity of 100%. No correlation was observed regarding patients’ sex, tumor staging and grading, and tumor site. Conclusion. The present study shows novel candidates that might be useful tools for oral cancer diagnosis. The neoexpression of Juno in cancerous tissues makes it a promising target molecule regarding its potential in diagnosis as well a therapeutic tool. Moreover, our observations suggest that also the repression of gene expression can be used for diagnosing—at least—OSCCs.

## 1. Introduction

Two decades ago, it was shown that three novel human embryonic genes including *Oct4* were re-expressed in cancer cells but not in somatic cells, suggesting these genes to be excellent candidates for cancer treatment [1]. Twenty years later, genes that are essential for embryonic development but nonessential for adult life were called a “goldmine” for digging cancer-specific targets [2]. Under this aspect, it is noteworthy that hundreds of embryonic germ line genes expressed by cancer cells that occur normally only during embryogenesis of reproductive cells have recently been identified. The idea behind this approach was to find more cancer-specific targets, which should not be expressed in any healthy tissue, and thus would be excellent targets for improved cancer treatment leading to fewer side effects [3]. A recently published study describes the use of ZP2, a prominent protein of human oocyte zona pellucida, as a new colon cancer biomarker [4]. A key protein in human tooth embryonic development, namely matrix-metalloproteinase 20 (MMP20), has been identified in various tumor entities, and is thus suggested to be useful as novel biomarker for cancer diagnosis [5]. In this context, the idea of using neoexpressed genes in tumor diagnosis is extended in this study with a larger number of various different embryonic genes to test their suitability as novel biomarkers that might lead to improved, unambiguous statements of diagnosis. 

Here, we describe our concept of establishing new biomarkers to improve oral tumor diagnosis. It is based upon the idea to search for specific genes that are known to be only expressed in distinct highly differentiated cells, e.g., oocytes or spermatozoa, but are neoexpressed in healthy or cancerous tissues on different transcript levels allowing an explicit and unambiguous differentiation between pathological and nonpathological conditions. We call this phenomenon “extorristic” gene neoexpression, derived from the Latin synonym for “homeless”. One example of this kind of genes, namely *matrix-metalloproteinase 20 (MMP20)*, has recently been published [5] showing a neoexpression of this proteolytic enzyme in various tumor tissues, although it has been known to mainly participate in the development of human teeth. Based upon two pilot studies that led to two patent applications, it turned out that a setup of various neoexpressed genes have been found to be expressed in a variety of different tumor cell lines, and thus might be relevant target molecules in tumor diagnosis. Hence, we were looking for genes that preferably allow a “yes” or “no” answer regarding their expression in respect to tissue specificity. That means that only genes have been chosen for investigation that either show a neoexpression in cancerous tissues with no detectable or different mRNA levels in the normal healthy counterpart or vice versa. Thus, we have selected the following extorristic genes considering a high potential for new biomarkers in oral tumor diagnosis:

Izumo 1 receptor is a surface protein anchored in the membrane of oocytes and absolutely crucial for gamete recognition and fertilization through interacting with its ligand “*Izumo1*” located on sperm surfaces. *Izumo 1 receptor* has also been identified to be *folate receptor 4* and renamed as “*Juno*” [6]. A “pubmed” [7] literature search revealed only publications dealing with Juno in respect to fertilization, thus supposing Juno to be exclusively expressed in female oocytes. Hence, this molecule seems to be a promising target molecule to serve as an efficient tumor biomarker.

Catsper is a calcium cation channel protein known to be exclusively expressed in spermatozoa and testis. It is essential for sperm motility and located in the membrane of the sperm flagellum. This ion channel consists of a protein complex containing four different α-subunits (Catsper 1–4) and at least three additional auxiliary polypeptides (Catsper β, γ, and δ) [8,9]. Since these genes are expressed under limited, exclusive conditions, i.e., only in male spermatozoa and testis, we consider Catsperβ and Catsperδ as promising target molecules for new reliable tumor biomarkers in OSCCs.

A second set of newly proposed biomarkers derives from the ferlin family. Ferlins are proteins which participate in membrane fusion processes, e.g., endo- and exocytosis. They are characterized by a number of C2 phospholipid-binding domains that are responsible for membrane–membrane interactions. The human ferlin family consists of six members. *Dysferlin* and *Myoferlin* show a temporal gene expression pattern in muscle development and differentiation. *Otoferlin* has been found to be expressed in the inner ear. *Fer-1-like-5 (Fer1L5*) is involved in mediating myoblast fusion, but has also been detected in pancreas [10,11]. Some members of ferlins have already been discovered in human cancer: *Dysferlin* is significantly associated with pancreatic cancer patient survival, while *Otoferlin* is correlated to renal clear carcinoma and Fer1L5 to lung adenocarcinomas. *Myoferlin* is highly expressed in pancreas, breast, kidney, and head and neck squamous cell carcinoma [11]. We have chosen these candidate genes as new tumor markers in oral cancer diagnosis because they show a distinct expression pattern and have already been shown to be involved in tumorigenesis, albeit not regarding OSCCs.

Another type of sperm receptors belongs to the oocyte zona pellucida glycoprotein family. Human zona pellucida matrix consists among others of four gylocoproteins, namely zona pellucida glycoprotein 1–4 (ZP1-4) [12]. Recently, *ZP2* has been extensively studied in colon cancer evaluating its potential function and usability as new biomarker [4]. Thus, this gene seems to be a good candidate to be examined in context with another tumor entities, e.g., OSCC. ZP3 has been described to be the primary sperm receptor [13]. This protein has already been detected in normal and cancerous tissues, particularly investigated in prostate cancer cells [14]. A treatment strategy in mice for ovarian cancer that is based upon immunization against murine ZP3 has also been published [15]. All in all, aware of these observations, we consider *ZP3* to be a potent candidate for OSCC diagnosis.

Our study describes a promising approach for characterizing oral malignancies with a new set of novel biomarkers using extorristic genes. We have focused on genes allowing a “yes” or “no” statement regarding their expression levels in healthy and malignant entities with sensitivities and specificities of 100% only. This study describes a new diagnostic alternative by using the above-mentioned novel biomarkers. For this approach, no mechanistic examinations are necessary and thus have not been performed. Our study focuses exclusively on differential expression analyses of these extorristic genes without making any statements regarding their putative mechanistic functions.

## 2. Materials and Methods

### 2.1. Tissue Sampling

Tissue biopsies were isolated during surgical procedures (*n* = 25 of each entity). Healthy and cancerous specimens were from different individuals. Procedures involving human tissue sampling followed a protocol approved by the ethical board of the University of Bonn (#067/18). All patients had been informed about the study and had signed a letter of informed consent. Tumor tissue selection based upon the following parameters: similar numbers of female/male patients, primary tumor, no recurrence, and no preoperative treatment (chemotherapy; radiation therapy).

### 2.2. Reverse Transcription Real-Time PCR

For the isolation of total RNA from oral tissue, the RNeasy^®^ Fibrous Tissue Mini Kit (Qiagen, Hilden, Germany) was used. In summary, 30 mg tissue was homogenized and lysed in RLT buffer using a homogenizer. Then, proteinase K solution was added to the lysate followed by an incubation step for 10 min at 55 °C. After transferring the proteinase K digested lysate to the RNeasy^®^ spin columns, an additional on-column DNA digestion was performed to remove potential DNA contaminations. Finally, total RNA was eluted from spin columns, analyzed for quantity and quality using a NanoDropHND-1000 spectrophotometer (NanoDrop Technologies, Wilmington, DE, USA), and stored at −80 °C. Reverse transcription of 1 µg total RNA was performed as described [16]. Transcript levels of all genes of interest were detected by real-time PCR using the CFX Connect™ Real-Time PCR System (Bio-Rad Laboratories, Munich, Germany), SYBR^®^ Green (Bio-Rad Laboratories), and specific primers (Table 1). All primers were subjected to verification by computational analysis for specification (BLAST) and synthesized of high quality (Metabion, Martinsried, Germany). Primer sequences, annealing temperatures, and efficiencies are shown in Table 1.

Quantitative PCR was carried out with 20 ng RNA equivalent per tube as previously described [16]. Relative differential gene expression was calculated using the ∆∆Ct-method described by Pfaffl [17] with β-actin, β-2-microglobulin (B2M), glyceraldehydephosphate-dehydrogenase (GAPDH), and ribosomal protein P0 (RP0) serving as housekeeping genes. Relative gene expression was standardized to the mean values for all reference genes. Real-time PCR was carried out according to “The MIQE Guidelines: *M*inimum *I*nformation for Publication of *Q*uantitative Real-Time PCR *E*xperiments” [18]. For absolute qPCR quantification, a known amount of numbers of molecules of ZP2 were used in dilution series. The limit of detection was determined at a cycle of quantification (Cq) value of 35 (=one molecule) (Figure 1) as described recently [4].

### 2.3. Statistical Analysis

Mean values and standard deviations of the samples (*n* = 25) were calculated. One-way ANOVA and the post-hoc Tukey’s multiple comparison test were applied using a statistical software program (GraphPad Software, San Diego, CA, USA). *p*-values less than 0.05 were considered to be statistically significant.

## 3. Results

In order to analyze low transcript levels, qPCR experiments with defined numbers of template molecules were performed to determine the detection limit. As shown in Figure 1, one single molecule could be detected at a Cq of “35” with ZP2 serving as representative template. The PCR efficiency exhibited linearity over eight logarithmic orders.

Relative expression profiling of neoexpressed genes of interest in tissue samples of oral mucosa and OSCCs was carried out to determine which genes were suitable to be used as new biomarker candidates (Table 2).

Transcripts coding for *CatsperB* were present as well in oral mucosa (0.001) as also in OSCCs (0.0005) with being nearly on the same level. This observation leads to the decision that CatsperB cannot be used as an acceptable biomarker. In contrast, *CatsperD* gene expression was detectable exclusively in oral mucosa (0.0015). Similar observations were found for samples of oral mucosa regarding *dysferlin* (0.0005), *Fer15L* (0.00085), and *otoferlin* (0.0036). All these four genes exhibited no detectable transcripts in OSCCs. Transcripts for *Myoferlin* and *ZP3* were positively detected as well in oral mucosa as also in OSCCs, with a higher expression rate for *Myoferlin* in oral mucosa (0.022) and OSCCs (0.0078) and low transcript levels for *ZP3* in oral mucosa (0.000024) and OSCCs (0.000081). The expression pattern of *Juno* was different in comparison to all other genes examined in our study: Juno was found to be present exclusively in OSCCs (0.0003) but fully absent in oral mucosa (Table 2).

Differential relative gene expression analyses exhibited a tenfold higher transcript level in cancerous tissues for Juno (Table 3).

In order to verify the suitability as a new biomarker candidate, sensitivity and specificity of the selected genes of interest were investigated (Table 4).

Comparative analyses of differential expression between oral mucosa and OSCCs (Table 3) revealed no significant difference of *CatsperB* with a 50% expression level of OSCCs (0.5) versus oral mucosa (1.0) although having a sensitivity and specificity of 100% (Table 4). In contrast, *CatsperD* showed a significant 50-fold higher expression rate in oral mucosa compared to OSCCs with no detectable amounts of mRNA (Table 2) but 100% sensitivity and specificity (Table 4). *Dysferlin* (16.7-fold) and *Fer5L5* (25-fold) also exhibited an enhanced transcript level in oral mucosa with no detectable gene products in OSCCs and showed also sensitivities and specificities of 100% (Table 3 and Table 4). A 2.9-fold enhanced gene expression rate of *Myoferlin* was detected in oral mucosa compared to OSCCs with 100% sensitivity and specificity (Table 3 and Table 4). Nevertheless, it is questionable if this gene can be supposed as a suitable biomarker for tumor diagnosis because of the low level of difference in expression. In contrast, *Otoferlin* exhibited a significantly highly increased transcript level by 111-fold in oral mucosa compared to nondetectable gene products in OSCCs and a sensitivity and specificity of 100% (Table 3 and Table 4). Gene expression of *ZP2* could be found in oral mucosa and also OSCCs with a 4.8-fold enhanced level in healthy tissues (Table 2 and Table 3). Nevertheless, *ZP2* transcripts were detectable in only 45% of oral mucosal tissue samples but above all in 38% of OSCC specimens (Table 4). Hence, this gene does not match our requirements as a reliable cancer-specific target. Finally, *ZP3* had a significantly 2.9-fold increased expression rate in oral mucosa compared to OSCCs (Table 3). While *ZP3* transcripts could be detected in all samples of oral mucosa, it was also traceable in 64% of OSCC samples (Table 4). This observation excludes *ZP3* as a suitable biomarker since our basis for decision of an appropriate diagnostic marker was 100% of sensitivity and specificity to gain an unambiguous “yes” or “no” answer.

In summary, from initially nine assumed putative target genes, only *Juno* seems to be a promising new biomarker for OSCCs. Nevertheless, the further four candidates (*CatsperD*, *Dysferlin*, *Fer1L5*, *Otoferlin*) could also be useful in characterizing oral malignancies.

For correlation analyses, all obtained qPCR results were examined with clinical data of the patients having participated in the present study. Details of clinical data from all patients examined are listed in Table 5.

(1). No sex-specific correlation could be found for all genes analyzed. (2). No correlation to age could be verified. (3). No correlation was observed, neither for tumor staging, nor for tumor grading according to the TNM-staging system [19]. (4). Regarding OSCCs, 56% were taken from patients suffering tongue cancer, whereas 44% were tumors from jaw. However, tumor site-specific correlations could not be drawn.

Hence, all above-mentioned useful candidates can be taken into account for biomarkers in tumor diagnosis, no matter which stage the tumor has developed.

## 4. Discussion

The objective of this study is the search for novel cancer-specific target genes as novel diagnostic alternative method for unambiguous tumor identification. Nine embryonic genes have been examined regarding their potential to give “yes” or “no” answers under the following assumption: Significant different gene expression in healthy versus cancerous tissues in connection with a sensitivity and specificity of 100%. As a result of this approach, particularly *Juno* as a real tumor marker but also four additional genes, namely *CatsperD*, *Dysferlin*, *Fer1L5*, and *Otoferlin* remain to be useful in characterizing oral malignancies. However, all investigated genes are included in the following discussion. For some genes (*CatsperB*, *CatsperD*, *Dysferlin*, *Fer1L5*, *Juno*), a limited number of published literature referring to cancer is known. For that reason, a database search in “genevisible” [20] has been performed to be able to properly discuss the results demonstrated in this work. *Juno* is the only gene in the present study that is absent in normal tissue but present in all of the examined tissue samples of OSCC. This characteristic defines *Juno* as an excellent target to serve as a cancer-specific marker. It is also the first time that *Juno* has been detected in tumor tissue samples. Further studies have to be performed to enlighten its putative function in OSCC. Additionally, Juno, as a cell membrane-bound receptor, might not only be a potential target for diagnosis but also for therapy. Database search in “pubmed” and “genevisible” has not revealed any information regarding *Juno* in respect to cancer or its expression besides in oocytes.

*CatsperB* was found to be deleterious in colorectal cancer, namely serrated polyposis syndrome [21]. A pancancer multiomics analysis identified CatsperB as a potential protein partner of Tac2-N [22]. “Genevisible” data analyses revealed *CatsperB* detectable in various human tissues including pancreas and colon and also in cancer of the prostate, pancreas, stomach, and esophagus. Thus, our results show for the first time *CatsperB* expression in oral mucosa and OSCC, although not suitable for diagnosing oral malignancies. However, *CatsperB* could be used as a cancer-specific target to identify prostate, colon, esophagus, and stomach cancer, since no expression has been detected in their healthy counterpart tissues [20]. *CatsperD* transcripts have been found primarily in nasal and airway epithelia and oocytes but not at all in cancer [20]. This is in agreement with our results showing *CatsperD* expression in only healthy tissue. Taken all this information together, *CatsperD* might be a suitable biomarker for identifying other types of cancer. In the present study, *Dysferlin* transcripts have been identified in healthy oral mucosa but not at all in OSCC tissue specimens. Data base analyses have revealed that *Dysferlin* is slightly downregulated in various cancer entities compared to their normal counterparts, including inter alia bladder urothelial carcinoma, breast invasive carcinoma, cervical carcinoma, cholangiocarcinoma, and colon and colorectal carcinoma, with OSCCs not incorporated in this search [11]. Using genevisible data bank, *Dysferlin* is mainly detected in various endothelial tissues, leukemia, and kidney cancer. In summary, *Dysferlin* can be used as target molecule for oral tumor diagnosis, perhaps even in other tumor entities as long as the score for specifities and sensitivities is high enough accompanied by significant differences on its expression level. Transcripts of another member of the ferlin family, namely *Fer1L5*, is also present in healthy oral tissues while not detectable in OSCCs. Peulen and coworkers have shown that this gene is also lightly downregulated in cancer of various further entities (e.g., lung, head and neck, stomach, esophagus) compared to their normal tissue counterparts [11]. Genevisible data search reveals mRNA presence in airway, nasal, and tracheal epithelia. *Fer1L5* can also be used as biomarker for diagnosis of OSCCs. A reliable prediction for its suitability for additional cancer entities is based on the same requirements applied for *Dysferlin*. In contrast to the genes mentioned above, *Myoferlin* as shown by various studies is indeed relevant concerning cancerous disease: it is generally up-regulated in tumors, e.g., in esophagal, renal, and glioblastoma with a strong correlation to survival of patients with glioma and pancreatic adenocarcinoma [11]. Although our results exhibited a slightly diminished expression rate in OSCC versus normal, yet, this observation is in good agreement with the results for other cancer entities (bladder, breast, cervix, prostate, rectum) [11]. While Myoferlin plays an important role in the degradation of epidermal growth factor receptor (EGFR) in breast cancer [23], loss of this gene activity leads to a change in cell motility, i.e., reduced velocity of cell migration [24]. High expression of *Myoferlin* is associated with poor patient outcome in oropharyngeal squamous cell carcinoma [25], and poor prognosis in pancreatic ductal adenocarcinoma [26], clear cell renal cell carcinoma [27], and colon cancer [28]. In contrast to these observations, high expression of *Myoferlin* in endometrioid carcinoma was connected to low-grade tumors, whereas loss of *Myoferlin* expression led to high-grade carcinomas, presumably because endometrial tissues are steadily undergoing a continuous process of regeneration [29]. This rationale could also apply for normal oral mucosal tissues which would explain the enhanced expression rate of *Myoferlin* compared to the cancerous specimens. Our study shows that *Otoferlin* is significantly higher expressed in normal tissue while nondetectable in OSCCs. Similar observations have been published for esophageal, pancreatic, and thymomal tumors [11]. In contrast, tumor entities regarding of bladder, breast, cervix, head and neck, and kidney origin exhibit opposite expression pattern, i.e., higher transcript levels in cancer versus normal [11]. Additionally, an increased *Otoferlin* transcript level correlates with poor prognosis in clear renal cell carcinoma [30]. Another gene of interest, namely *ZP2*, has recently been identified as new target molecule in colon cancer. This study also shows ZP2 presence in cell culture of various tumor origin [4]. In addition, ZP2 protein has also been found in prostate cancer and corresponding PC3 cells, too, whereas ZP2 mRNA could, however, not be detected [14]. In our present study, *ZP2* shows a specificity and sensitivity below 100% which excludes this gene as potential cancer-specific marker. Nevertheless, these observations are yet in agreement with our recent study on colon cancer [4]. Finally, another member of the zona pellucida glycoprotein family, namely *ZP3*, has been chosen for examination. Although *ZP3* expression has been found in 100% of normal tissue, it has also been present in 64% of OSCC samples which leads to an exclusion of *ZP3* for a good potential cancer-specific biomarker. However, in mice, ZP3 has been used as treatment approach against ovarian cancer based upon immunization against murine ZP3 [15]. A recent publication deals with *ZP3* as new prognostic and potential therapeutic marker in renal clear cell carcinoma showing *ZP3* 30-fold higher expressed in tumor versus normal counterparts [31]. This observation is in contrast to our results.

In summary, our study shows a number of novel candidates as useful additional methods for diagnosis of oral cancer. Particularly, Juno, as a cell-membrane receptor, seems to be an interesting gene regarding its potential as well as a diagnostic and also as a therapeutic alternative. Besides that, it is the first time to our knowledge that Juno has been detected in a tissue other than oocytes.

The absence of functional analyses might be a limitation of the present study. However, regarding *Juno*, no function other than binding Izumo1 has been published up to date [6]. Nevertheless, we have analyzed Izumo1 structure motifs responsible for binding Juno [32] for putative similar proteins with the “blast” program but found exclusively Izumo1 as a search result. Furthermore, we have performed a sequence analysis for Juno using “blast” for putative other candidates to get an indication for other functions than binding to Izumo1. We have even sliced Juno sequence into eight fragments and carried out sequence analyses with no other results but Juno. Hence, there is still no indication for a putative tumor associated function for *Juno*. Yet, our approach does not require mechanistic examinations, and thus none were carried out. The present study focuses exclusively on differential expression analyses of the above-mentioned extorristic genes without making any statements regarding their putative mechanistic functions.

## Figures and Tables

**Figure 1 jpm-12-00494-f001:**
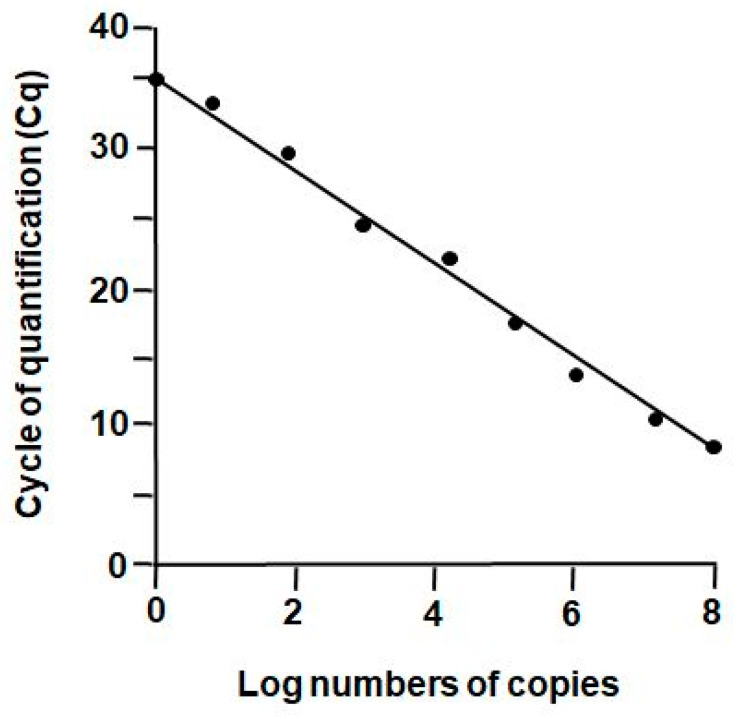
Standard curve for *ZP2* detection using dilution series with known copy numbers. The detection limit is at Cq = 35 for one molecule positively detected.

**Table 1 jpm-12-00494-t001:** Genes, primer sequences, corresponding efficiencies, and annealing temperatures.

Gene	Primer Sequences (Sense/Antisense)	Efficiency	Annealing Temperature (°C)
β-actin	5′-CATGGATGATGATATCGCCGCG-3′ 5′-ACATGATCTGGGTCATCTTCTCG-3′	1.84	69
B2M	5′-GCCTTAGCTGTGCTCGCGCT-3′ 5′-TGCTGCTTACATGTCTCGATCCCA-3′	2.04	64
GAPDH	5′- TGGTATCGTGGAAGGACTCA-3′ 5′-CCAGTAGAGGCAGGGATGAT-3′	1.93	67
RPO	5′-GCCTTGACCTTTTCAGCAAG-3′ 5′-GCAGCATCTACAACCCTGAAG-3′	1.97	62
CatsperB	5′-TCTTTTTGGACAGCCTCCAGATATGGG-3′ 5′-ACAAGGCCTGAACAACTTGTGAATCA-3′	2.02	62
CatsperD	5′-GGTGGAGCTGTGGCGAAAAGAC-3′ 5′-CTCCAGTTGACAGTAGCTGTAGTACGG-3′	2.08	57
Dysferlin	5′-TGGTGGTCAAAGACCATGAG-3′ 5′-ACATCCAGGTCAGGCAGAGT-3′	1.89	55
Fer1L5	5′-CGCTTGTCGGGCAAGGTGAAG-3′ 5′-GCCAAATAGTGCGGAGCTGAATAGATG-3′	1.94	56
Juno	5′-CTCTATGAGGAGTGCATCCCCTG-3′ 5′-CGGTTCTTCCCCTGACTCCAG-3′	2.06	60
Myoferlin	5′-TGTGGAATCTGCCAGCAATA-3′ 5′-CAGGTCCTTCAGGGCTACAG-3′	1.88	62
Otoferlin	5′-ATGGCCACCGGGGAGGTGGA-3′ 5′-AGCTCGTGTCGGGCCGGTTG-3′	2.01	63
ZP2	5′-GCTCTCTAGCCTGGTCTACTTCCACT -3′ 5′-GTCCATAGCACCTCGTGAGCCA-3′	2.08	69
ZP3	5′GGATGTGTCCGGTGCCATAG-3′ 5′-CACTCGTGGAGTCCAACCTC-3′	2.05	61

**Table 2 jpm-12-00494-t002:** Relative expression of biomarker candidates in samples of oral mucosa and oral squamous cell carcinomas (OSCC) (*n* = 25).

	Oral Mucosa	OSCC
Juno	n.d.	0.0003 (0.00007)
CatsperB	0.001 (0.0002)	0.0005 (0.0001)
CatsperD	0.0015 (0.0004)	n.d.
Dysferlin	0.0005 (0.00009)	n.d.
Fer1L5	0.00085 (0.00014)	n.d.
Myoferlin	0.022 (0.006)	0.0078 (0.002)
Otoferlin	0.0036 (0.0007)	n.d.
ZP2	0.0000047 (0.00000073)	0.000001 (0.00000004)
ZP3	0.000024 (0.000007)	0.000081 (0.000023)

Mean values are shown with standard deviations in brackets. “Not detectable” is abbreviated as “n.d.”.

**Table 3 jpm-12-00494-t003:** Differential relative gene expression analysis in samples of oral squamous cell carcinomas (OSCC) as xfold compared to oral mucosa.

	Oral Mucosa	OSCC
Juno	n.d.	10.0 *
CatsperB	1.0	0.5
CatsperD	50.0 *	n.d.
Dysferlin	16.7 *	n.d.
Fer1L5	25.0 *	n.d.
Myoferlin	1.0	0.35 *
Otoferlin	111.0 *	n.d.
ZP2	1.0	0.21 *
ZP3	1.0	0.34 *

“Not detectable” is depicted as “n.d.”. A Cq-value of “36” was put into calculations to determine differential expression rate factors for samples with nondetectable gene transcripts according to Figure 1. Significant differences are marked with asterisks (* for *p* < 0.05).

**Table 4 jpm-12-00494-t004:** Descriptive expression analysis of biomarker candidate genes in tissue samples of OSCCs and healthy oral mucosa.

	Oral Mucosa		OSCC	
	+	−	+	−
Juno	0	100	100	0
CatsperB	100	0	100	0
CatsperD	100	0	0	100
Dysferlin	100	0	0	100
Fer1L5	100	0	0	100
Myoferlin	100	0	100	0
Otoferlin	100	0	0	100
ZP2	45	55	38	62
ZP3	100	0	64	36

Data show specificity and sensitivity (in percentage (%)) of detected (“+”) or nondetectable (“−“) neoexpressed genes.

**Table 5 jpm-12-00494-t005:** Sex, age, tumor staging and grading, and tumor site of patients examined in the present study.

Patient	Sex	Age	Staging/Grading	Tumor Site
1	m	70	pT2, N0, M0, L0, V0, Pn0, R0, G1	jaw
2	f	69	pT2, N0, M0, L0, V0, Pn0, R0, G2	tongue
3	m	67	pT2, N2b, M0, L0, V0, Pn1, R0, G1	jaw
4 (†)	m	64	pT2, N2c, M0, L1, V0, Pn0, R0, G2	tongue
5	m	72	pT2, N0, M0, L1, V0, Pn0, R0, G2	jaw
6 (†)	m	69	pT2, N2c, M0, L1, V0, Pn1, R0, G3	jaw
7	f	72	pT1, N0, M0, L0, V0, Pn0, R0, G2	tongue
8	m	63	pT1, N0, M0, L0, V0, Pn0, R0, G1	tongue
9	m	59	pT2, N0, M0, L0, V0, Pn0, R0, G2	tongue
10 (†)	m	60	pT2, N0, M0, L0, V0, Pn0, R1, G3-4	jaw
11	m	73	pT2, N0, M0, L0, V0, Pn1, R0, G2	tongue
12	m	78	pT2b, N2b, M0, L0, V0, Pn0, R0, G2	tongue
13	f	60	pT3, N0, M0, L0, V0, Pn0, R0, G2	tongue
14 (†)	f	82	pT4a, N2c, M0, L0, V0, Pn0, R0, G2-3	jaw
15	f	62	pT1, N0, M0, L0, V0, Pn0, R0, G1	tongue
16 (†)	m	96	pT3, Nx, M0, L0, V0, Pn1, R0, G2	jaw
17	f	60	pT2, N1, M0, L0, V0, Pn0, R0, G2	tongue
18	m	65	pT2, N0, M0, L0, V0, Pn0, R0, G2	tongue
19	m	47	pT1, N0, M0, L0, V0, Pn0, R0, G2	tongue
20	m	70	pT2, N1, M0, L0, V0, Pn0, R0, G2	tongue
21	f	69	pT2, N0, M0, L0, V0, Pn0, R0, G2	jaw
22	m	62	pT3, N0, M0, L0, V0, Pn0, R0, G3	jaw
23 (†)	f	98	pT4a, N2b, M0, L0, V1, Pn1, R1, G2	jaw
24 (†)	m	62	pT4a, N0, M0, L0, V0, Pn0, R0, G3	jaw
25	m	78	pT1, N0, M0, L0, V0, Pn0, R0, G1	tongue

Patients with † died in the meantime. Staging/grading is abbreviated according to the TNM-staging system. Abbrev.: T—Tumor; N—Node; M—Metastasis; L—Invasion of lymphatic vessels; V—invasion of vein; Pn—invasion of nerves; R—completeness of resection; G—Grading.

## Data Availability

Supporting data and materials are available on request to the corresponding author.

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
