# Peer review of "Neoexpression of JUNO in Oral Tumors Is Accompanied with the Complete Suppression of Four Other Genes and Suggests the Application of New Biomarker Tools"

_jpm, 2022, doi:10.3390/jpm12030494_

Round 1
Reviewer 1 Report
The manuscript needs to be improved for
- The co-relation analysis has to be in detail, especially line 223-226 is not accurate.
- The entire writing needs to be formatted for grammar errors. Many sentences are not scientifically accurate like line 201" Myoferlin was higher expressed in oral mucosa"
- Discussion needs to be more elaborated especially regarding the functions of Juno and other genes identified and their implication in OC and precancers.
- Limitations of the study is not addressed.it has to be addressed as a single paragraph before the conclusion
- The line 114 is not accurate. It must be changed. There is no tool involved.
Author Response
Comments to reviewer
Dear reviewer,
we are very grateful for all your comments and suggestions which we really appreciate. They were very helpful and have definitely improved our manuscript.
- We have changed the section (lines 223-226) according to your remarks. Now, lines 205ff are located behind Table 4.
- We have changed the part (lines 201ff) according to your suggestions. Now lines 212ff.
- We have carried out blast analyses for the Juno binding motifs of Izumo1 but have found exclusively Izumo1. We have added this observation at the end of „discussion“ including a new reference [32]. Furthermore, we have performed a blast for Juno for putative other candidates to get an indication for other functions than binding to Izumo1. We have even sliced Juno sequence in 8 fragments and carried out blast with no other result but Juno. Thus, we cannot even speculate about a putative function of Juno in tumor cells.
- We have added a paragraph at the end of the manuscript.
- We have changed line 114 (now line 117).
General:
We have changed the order in „results“ a bit to improve readability regarding intersections between Tables.
All changes are in „red“.
We have now corrected typing errors and spacing errors. The manuscript has also been edited for English language.
Reviewer 2 Report
Thanks to the introduction to the increasingly frequent use of quantitative PCR gene amplification tests and the availability of targeted gene probes, the study of the typing of neoplastic tissues is experiencing a significant increase. All the data collected can be the basis of a targeted therapy, as well as providing useful information on the genetic mechanisms that underlie the behaviour of tumors.
In the oral cavity, these data are certainly less numerous and derived from small series.
The work is set up and developed correctly both in the methodological and in the statistical part.
The value of the data obtained on the JUNO gene sequence is however of a descriptive nature, as clearly stated by the authors themselves.
The English form should be perfected by synthesizing some sentences as well as by using less colloquial expressions.
In the materials and methods, the inclusion criteria must be specified, such as data on previous treatments, on second tumors in the body, if relapses after surgery and / or radiotherapy etc.
More important is to describe in table 5 the sub-site of the tumor, the classification used (TNM 8 ed is assumed) and in the legend describe the abbreviations used to report the adverse factors.
Author Response
Comments to reviewer
Dear reviewer,
we are very grateful for all your comments and suggestions which we really appreciate. They were very helpful and have definitely improved our manuscript.
- We have added inclusion criteria in „materials&methods
- We have added sub-sites of tumors and abbreviations according to „TNM 8th ed.“ (also changed in reference 19).
General:
We have changed the order in „results“ a bit to improve readability regarding intersections between Tables.
All changes are in „red“.
We have now corrected typing errors and spacing errors. The manuscript has also been edited for English language.
Round 2
Reviewer 1 Report
Nil